# In-Patient Multimodal Intensive Neurorehabilitation and Care Improve Motor and Non-Motor Functions in the Moderately Advanced Stages of Parkinson’s Disease: A Retrospective Study in a U.S. Facility

**DOI:** 10.3390/biomedicines12081658

**Published:** 2024-07-25

**Authors:** Priyanka Moondra, Lyubov Rubin, Mara McCrossin, Amanda Persaud, Alessandro Di Rocco, Angelo Quartarone, Maria Felice Ghilardi

**Affiliations:** 1Department of Neurology, Movement Disorders Division, Northwell Health, Donald and Barbara Zucker School of Medicine at Hofstra/Northwell, Long Island, NY 11542, USA; moondrap@gmail.com (P.M.); lrubin3@northwell.edu (L.R.); mmccrossin@northwell.edu (M.M.); apersaud20@northwell.edu (A.P.); adirocco1@northwell.edu (A.D.R.); 2IRCCS Centro Neurolesi “Bonino-Pulejo”, 98124 Messina, Italy; 3Department of Molecular, Cellular and Biomedical Sciences, CUNY School of Medicine, New York, NY 10031, USA

**Keywords:** neurorehabilitation, exercise, motor symptoms, non-motor symptoms, Parkinson’s disease, intensive care

## Abstract

(1) Background: Previous studies, mostly performed in European centers, have shown that in-patient multimodal intensive rehabilitation treatments lasting for two to four weeks can improve both motor and non-motor symptoms of Parkinson’s disease (PD) with long-lasting effects. Here, we ascertain the effects of a similar in-patient program in a U.S. center with a retrospective study in a cohort of 153 patients in the moderately advanced stage of PD. (2) Methods: We compared indices of motor and non-motor functions before and immediately after such treatment and investigated the possible differences between men and women. We used the available records of the Beck Depression Inventory, PDQ39, PD Sleep Scale, Timed Up and Go, Vocal Volume, Voice Handicap, and total UPDRS scores. (3) Results: We found that at the end of treatment, which lasted an average of 14 days, all outcome measures significantly improved independently of sex. (4) Conclusions: These results confirm the previous findings with a similar in-patient approach in European centers. They further suggest that this in-patient treatment is a care model that is feasible in U.S. centers and can provide a more immediate benefit to the motor function and quality of life of patients with moderately advanced PD.

## 1. Introduction

Parkinson’s disease (PD) is a progressive neurodegenerative condition manifesting with motor and non-motor symptoms. Together with tremors and/or rigidity, bradykinesia is the mainstay of a PD diagnosis [1], although mood dysregulation, cognitive abnormalities, pain, autonomic dysfunction, and sleep problems significantly contribute to the disabling aspects of this disease [2,3]. The global prevalence of PD has doubled in the last 25 years and may rise further as the population ages [4]. The projected future economic burden will also pose challenges to the healthcare system at large, necessitating the development of more cost-effective care [5]. Pharmacotherapy can control, at least in part and only for some time, most of the symptoms but does not slow the progression of disease. Surgical interventions may not be available for many patients, including those in the advanced stages, with cognitive impairment or older age. For all these reasons, the integration of complementary approaches into conventional treatment regimens has become an important staple of care throughout the disease course and especially in the early stages. In particular, neurorehabilitation plays an important part of the comprehensive care regimen to combat the motor decline in PD [6]. In combination, physical therapy (PT), occupational therapy (OT), and speech language therapy (SLT) may help in preserving the functional abilities of patients for longer. Meta-analyses of physiotherapy trials for PD have found that conventional methods significantly improved balance, gait, and quality of life [7,8]. The results of many studies in both animal and humans also supports the inclusion of aerobic exercise as a neuroprotective treatment strategy in programs for the care of PD [9]. The positive effects of exercise on motor and non-motor symptoms may be mediated throughout various mechanisms, including the enhancement of plasticity, increased availability of dopamine, inhibition of oxidative stress, and increased clearance of alpha-synuclein [10,11,12,13,14]. Despite many studies, there are no official guidelines defining the precise modalities, dosing, and settings for rehabilitation interventions in PD. Nevertheless, in-patient programs based on a multimodal rehabilitative approach with an important aerobic component has shown efficacy at all stages of the disease for both motor and non-motor symptoms [15,16,17]. In particular, the results of European randomized clinical trials have shown that such programs may slow down the progression of the disease in the early PD stages [16] and can also be efficacious for moderately advanced patients at a higher risk of falling and losing independent function [18,19,20,21,22,23]. In general, these multimodal programs can lead to stability of activity of daily living and delay nursing home admissions [24], with a significant impact on the quality of life of both the patients and caregivers.

Despite the evidence coming from randomized clinical trials, such in-patient multimodal intensive rehabilitation programs have not been yet established in the United States because of several challenges including reimbursement problems. Here, we present the preliminary results of a retrospective study about the acute effects of an in-patient Multimodal Intensive Neurorehabilitation and Care (iMINC) protocol initiated at Glen Cove Hospital in 2021. This protocol included PT, OT, SLT, recreational and art therapy, neuropsychology, and dietary assessment with interventions of social workers, pharmacists, nurses, and neurologists. We thus reviewed the charts of over 150 patients with advanced PD who participated and completed the program at Glen Cove Hospital. We particularly focused on the collected measures of general well-being, motor and vocal function, depression, and sleep. We further investigated whether such an in-patient multimodal intensive approach brought similar benefits to men and women affected by the disease. Importantly, retrospective studies, such as the present one, are essential in clinical research as a step to identify patterns or relationships to guide prospective randomized, controlled trials. Indeed, a priori identification of possible biases or constraints and the utilization of existing data allow for significant savings of both time and economical resources. Moreover, retrospective studies also allow us to capture real-life scenarios and examine the outcomes as they occur naturally in real-world settings, outside the controlled environment of a randomized, controlled trial. An initial validation of the European results of iMINC in the U.S. system is warranted because of the differences in socio-economical context and type of healthcare system. Thus, the retrospective nature of this study provides a rapid and low-cost way to attain a preliminary confirmation of the European data, thus building the bases for future randomized controlled trials in the U.S.

## 2. Materials and Methods

### 2.1. Study Design

This retrospective study is based on the review of the charts of 153 patients with PD (Hoehn and Yahr Stage 3–4) that were pre-approved for elective admission to the Parkinson’s Neurorehabilitation Unit at Glen Cove Hospital by Medicare or commercial payors in the period from May 2021 to February 2023. Criteria for admission to the iMINC program at Glen Cove included the following: loss of independence in activities of daily life; mobility and gait impairment with recent falls; speech and swallow impairment; preserved cognitive abilities; and failure to achieve meaningful improvement with outpatient therapy.

### 2.2. Intervention

Patients were enrolled in a structured multidisciplinary program with an emphasis on physical, occupational, and speech therapies. A neurorehabilitation specialist was the primary provider, while movement disorder specialists monitored the patient and tailored pharmacotherapy during the enrolment period, and the patient was also medically managed by a hospitalist throughout the course of the stay. An intensive approach was taken for therapies, with three hours of individualized PT, OT, and SLP offered 5 days a week and an additional two hours of group therapy. The duration of each session, including warm up and recovery periods, was about one hour for each discipline and adjusted based on the patients’ needs.

Table 1 summarizes the different therapies and activities for each session. The focus of these therapies was multifactorial. In PT sessions, exercises included the following: treadmill walking, stationary cycling, and elliptical use together with stretching and range of motion exercises to improve movement of spinal, pelvic, and scapular joints; strength and resistance training with weights and bands were used to improve functionality of core muscles and to help with gait and posture; and balance exercises with cueing, such as metronomes and music, were also utilized. OT sessions were tailored according to the patients’ needs. In general, the focus of OT sessions was to enhance autonomy in activities of daily living with exercises to improve proprioception, hand functionality, ability of transfers out of bed, ability in dressing, and self-care. Dynamic exercises such as manipulation of tools, sewing, or stringing beads were implemented to help with hand movements. Handwriting or typing exercises were also used. SLT utilized the Speak-Out program to improve vocal volume, intensity, and patients’ ability to communicate effectively. Swallow and dysphagia therapy was also included based on patient evaluation and need. Group therapies made up the rest of the time, for a total therapy time of 5 h per week. These therapies included painting, creative arts, group games, and Dance for PD programming. They targeted mood, socialization, and movement.

### 2.3. Outcome Measures

A variety of clinical and self-reported scales were administered by trained therapists and/or the clinical team at admission, before the intervention, and at the end of the iMINC. Specifically, we collected the scores of the following: Beck’s Depression Inventory (BDI) in 153 patients, PD Sleep Scale (PDSS) in 153 patients, Timed Up and Go (TUG) in 151 patients, Vocal Volume in 151 patients, Voice Handicap in 124 patients, Parkinson’s Disease Questionnaire (PDQ-39) in 151 patients, and the total score of the Movement Disorder Society-Sponsored Revision of the Unified Parkinson’s Disease Rating Scale (MDS-UPDRS) in 79 patients.

### 2.4. Data Analysis

To analyze the intervention-related changes of all the outcome measures, we used mixed-model ANOVA with time (pre- and post-treatment) and sex (men and women) as the between-subject factors. When appropriate, post hoc tests were performed. The threshold of *p* value significance was set at 0.05. Bayes Factor approximation was also considered by computing the Vovk–Sellke maximum p-ratio (VS-MPR) [25], a function of *p*-value that represented the lower bound of the Bayes Factor—favoring H0 to H1—for a wide range of different prior distributions. This also ensured that the absolute probability of a particular model would be a good explanation of the observed data. VS-MPR values greater than 2.46 were considered significant. We also verified effect size by computing partial eta-squared (η² p), i.e., the % of the variance in the dependent variable attributable to a particular independent variable. Differences between men and women in terms of age and length of stay were analyzed with Welch tests for unequal samples with *p* value significance set at 0.05 and VS-MPR at 2.46. We also determined the effect size with Cohen’s d values and 95% confidence intervals (CI). All statistical analyses were performed using JASP (version 0.14.1).

## 3. Results

A total of 153 subjects with PD (90 men, median age 75 years) underwent the iMINC program at Glen Cove Hospital for a minimum of 6 days to a maximum of 27 days, with a median of 14 days.

There were no differences between men and women in terms of both age (Welch test (132.2) = −0.43, *p* = 0.669, VS-MPR = 1, Cohen’s d = 0.071, 95% CI: −0.39, 0.25) and length of stay (Welch test (126.0) = 0.85, *p* = 0.396, VS-MPR = 1, Cohen’s d = 0.141, 95% CI: −0.46, 0.18). The mean and standard deviation of age and length of stay as well as those of the outcome measures before (Pre) and after (Post) the iMINC program are reported separately for men and women in Table 2.

The results of the mixed-model ANOVA analyzing treatment-related changes in men and women are reported in Table 3. Briefly, they showed significant improvements after the treatment for all the outcome measures. Such effects were similar for all measures except for voice volume, as shown by the significant interaction between time and sex. Indeed, voice volume improvement was slightly but significantly greater in men (mean change ± SD: 5.93 ± 3.6) compared to women (4.53 ± 4.17, Welch test (116.2) = 2.13, *p* = 0.036, VS-MPR = 3.1, Cohen’s d = 0.36, 95% CI: 0.028, 0.685). Notably, the PDQ39 values were lower in men than in women both pre- and post-iMINC, as suggested by inspection of the values reported in Table 1, by the significant effect of sex (see Table 2) and by the post hoc tests comparing men and women before (*t* = −2.995, *p* = 0.007) and after the treatment (*t* = −3.237, *p* = 0.006). Nevertheless, treatment had the same effect on PDQ39 in men (mean change ± SD: 5.78 ± 14.38) and women (5.11 ± 11.03, Welch test (149.7) =−0.33, *p* = 0.74, VS-MPR = 1, Cohen’s d = 0.05, 95% CI: −0.375, 0.269).

Altogether, these findings suggest that the iMINC program is feasible and effective in improving motor and non-motor functions for persons affected by moderately advanced PD, independently of sex.

## 4. Discussion

The present findings demonstrate that at the end of a multimodal intensive neurorehabilitation and care program, there is a global improvement in the functional status of patients with moderately advanced PD. These results are the first ones coming from a U.S. center and provide an initial validation in the U.S. of previous European findings, mostly from randomized controlled trials and other studies [16,17,18,19,20,21,22,23,26,27]. Despite the limitations inherent in the retrospective nature of the present study, this initial validation of European results in the U.S. system is a rather crucial (and not foregone nor trivial) step, as the impact of any therapeutical approach or complex strategy is strongly dependent on the type of health system as well as on cultural and social differences. Thus, these results, which come at low-cost, provide a solid base for future randomized, controlled trials with follow-up testing in the U.S. focused on the efficacy of iMINC in PD.

We found that motor and non-motor performance and symptoms improved at the end of iMINC, as shown by TUG decreases, vocal performance amelioration, BDI score decreases, and UPDRS total score reductions. The improvements in all these domains are not surprising, as iMINC directly addresses most of the typical problems of PD, such as gait, balance, speech, and rigidity on the motor side, and depression and cognitive impairment on the non-motor side. Indeed, intensive, repetitive, and challenging goal-based practices are considered as rather important contributors to improve outcome measures that are directly related to the specific tasks practiced.

Aside from improvements in areas that were the targets of the goal-based practice, our study revealed general improvements in the quality of life (i.e., PDQ39) as well as beneficial effects for sleep (PDSS scores), in agreement with previous findings of a controlled European study in PD patients testing the effects of intensive multidisciplinary rehabilitation [16]. It is possible that sleep improvements reflect the enhancement of neuroplasticity [28], a feature that is greatly diminished in PD [29,30]. The enhancement of neuroplasticity may be related to aerobic exercise, a mainstay of iMINC. Indeed, the goal-based tasks (i.e., training for gait and balance, active and passive exercises for joints’ mobility, stretching, strengthening of paravertebral muscles, and occupational and speech therapy psychological support) were accompanied by daily aerobic exercise. The high levels of daily intensive or aerobic exercise of iMINC may ameliorate oxygen consumption, cardiovascular and respiratory efficiency, as well as skill formation, likely through increases in the BDNF expression [31], promotion of BDNF and TrkB interaction, and by lowering the threshold for LTP induction [11]. Moreover, intensive exercise may reduce neuroinflammation [32] and oxidative stress [12], activate neurotrophin-signaling pathways [13], angiogenesis [33], and neurogenesis [34], while decreasing the levels of alpha-synuclein [14]. The possible effects of aerobic exercise on plasticity-related phenomena are also supported by findings of increases in the BDNF serum levels and BDNF-trkB activation in the lymphocytes [35,36] of patients with PD following an inpatient multimodal intensive rehabilitation program. An important iMINC feature that can be relevant for plasticity induction and distinguished from most rehabilitation approaches is the intensity of this program. Most programs for PD or other conditions are based on 45 min to 1 hour of treatment/day, three times/week for 4–8 weeks, for a total treatment time of 12–24 h. Instead, iMINC, which was based on at least four hours/day, five days/week for two weeks, provides a total of 40 h of personal treatment that may be considered as a high-dose therapy. Such a short but intensive approach can be more effective than less intensive long-duration programs because it provides a boost to enhance and efficiently engage long-term plasticity processes that are reduced in PD. Indeed, in PD, enhancements of plasticity indices can already be present after two weeks of intensive rehabilitation treatment [35]. Nevertheless, the precise mechanisms at the base of the present findings need a better definition.

Another important feature of iMINC is the possibility to review drug treatment, its dosage and schedule at the beginning of the treatment without the constraints of outpatient office visits, and to eventually review it during the patient stay. While the therapy changes pre- and post-iMINC were not significant, it is very likely that the therapy adjustments occurring before the treatment initiation played an important role in the patients’ improvements. In addition, aerobic exercise may lead to a reduced need for dopaminergic drugs, as animal models showed that dopamine availability increases with aerobic exercise [10].

In summary, all the components of iMINC probably contributed to the beneficial effects observed in the patients as follows: While goal-directed tasks may engage specific pathways that are affected by PD, aerobic exercise may provide a fundamental background for increasing plasticity and maintaining the effects of training; the environment and the care may boost reward system further enhancing plasticity and positive effects; increased attention to drug therapy may further improve the outcome. It is likely that synergy between all these interventions may produce a greater impact than the sum of the effects of the iMINC’s individual components.

A final consideration is that iMINC and its derivatives can provide an important tool not only to decrease the burdens of patients and caregivers, but also to reduce costs to society at large. In fact, this kind of integrated program can lead to a general reduction in the number of hospitalizations and falls [27,37]. Extending this intervention to patients in the early stages of the disease may further increase savings, as European studies have found that intensive rehabilitation may be particularly effective in the earlier disease stages [27] and that early interventions may delay disease progression [26]. Moreover, earlier exposure to this treatment may serve to motivate and educate patients to maintain a higher level of exercise at home and later in their life.

There are many limitations in this investigation. As previously and extensively discussed, this is a retrospective study, solely based on chart review, without any control group. In addition, it is not clear whether the beneficial effects could be maintained for a long time. A crucial aspect for a broader implementation of the program is its cost-effectiveness, an aspect that can be settled only with long-term follow-up studies. So far, this two-week program has been approved and paid for by Medicare. Additional advantages include direct (a two-week respite period) and indirect (patients’ improvements) benefits for the caregivers. We also expect that our follow-up studies with a control group will still yield similar improvements, as germane European programs showed disease progression slowdown in the early stages [16] and increased independent functioning in the later stages [18,19,20,21,22,23,24]. While the relatively large number of subjects in the present cohort of patients with advanced PD allowed us to consider sex as a possible factor, only a prospective study designed with the appropriate control groups and tested in a standardized way will enable us to answer the questions generated by the results of this study. For instance, would a similar integrated care approach limited to an outpatient setting yield similar results? While such a comparison is not currently possible, a recent European study compared the effects of inpatient treatments similar to iMINC to those of programs where patients managed by expert neurologists participated in outpatient enriched treatments. The results showed that outpatient programs produced a better motor outcome over an almost two-year time frame, improved quality of life, and reduced the number of falls and hospitalization [27]. Other questions include the following: Which are the specific contributions of each component of the iMINC program? Which of the components or combination is more relevant? What is the right amount of aerobic exercise? What is the minimum treatment duration required to obtain significant long-lasting results? Are these effects long lasting? Can these benefits be maintained with some home- or community-based programs? How frequently should a patient undergo in-patient treatments to maintain beneficial effects? Is this intervention attractive for patients coming from culturally diverse communities?

Despite the weaknesses and the open questions, the results of this retrospective analysis provide a preliminary proof of concept that this model is a feasible and reimbursable intervention in the U.S. and can produce subjective and objective benefits that replicate those reported in previous controlled European studies.

## Figures and Tables

**Table 1 biomedicines-12-01658-t001:** Daily activities during iMINC.

*1.* *Amplitude training* ○ LSVT Speak Out!
*2.* *Reciprocal patterns and aerobic exercises* ○ Treadmill walking ○ Stationary cycling ○ Elliptical machines *3.* *Balance therapy and gate training*
*4.* *External cued therapy* ○ Visual cues ○ Verbal Cues ○ Rhythmic auditory cues (metronome, chanting, music)
*5.* *Task-specific training*
*6.* *Community-based exercises*
*7.* *Resistance training with muscle strengthening and stretching*

**Table 2 biomedicines-12-01658-t002:** Patients’ characteristics. Means and standard deviations of age, length of stay, and outcome measures before (Pre) and after (Post) iMINC program are reported for men and women. N: number of subjects for each test. BDI: Beck Depression Inventory; TUG: Timed Up and Go; UPDRS Total: Total score of the Movement Disorder Society-Sponsored Revision of the Unified Parkinson’s Disease Rating Scale (UPDRS).

		Men			Women		
		N	Mean	SD	N	Mean	SD
Age (years)		90	74.04	9.65	63	74.73	9.80
Stay (days)		90	15.00	3.55	63	15.52	3.88
BDI	*Pre*	90	15.56	9.11	63	16.94	9.69
	*Post*		10.23	7.67		12.87	8.19
PDQ39	*Pre*	90	40.19	16.71	63	48.59	18.16
	*Post*		34.41	16.69		43.48	16.98
PD Sleep Scale	*Pre*	90	87.64	27.27	63	93.91	23.27
	*Post*		109.48	21.07		107.68	22.25
TUG (ms)	*Pre*	90	48.64	33.94	61	59.57	41.34
	*Post*		37.48	31.09		41.62	25.78
Vocal Volume	*Pre*	90	54.02	4.96	61	54.64	5.03
	*Post*		59.95	5.23		59.18	5.50
Voice Handicap	*Pre*	71	52.11	26.52	53	42.55	27.09
	*Post*		43.59	26.01		38.90	24.53
UPDRS Total	*Pre*	49	114.04	25.96	30	114.53	31.83
	*Post*		72.92	25.96		73.83	25.13

**Table 3 biomedicines-12-01658-t003:** Results of mixed-model ANOVAs for the outcome measures for the data in Table 1. Effects of the iMINC are reported with within-subject effects (time: Pre vs. Post-iMINC) and between-subject effects (sex: men, women). Significant results (i.e., *p*-values < 0.05 and VS-MPR > 2.46) are reported in bold. *p*: *p* values; η² p: Partial Eta Squared; VS-MPR: Vovk–Sellke Maximum Peak Ratio.

		BDI	PDQ39	PD Sleep Scale	TUG	Vocal Volume	Voice Handicap	UPDRS Total
Time	F	**65.96**	**25.57**	**72.83**	**51.14**	**269.46**	**9.65**	**186.24**
(df:1)	*p*	**1.51 × 10^−13^**	**1.22 × 10^−6^**	**1.39 × 10^−14^**	**3.60 × 10^−11^**	**3.15 × 10^−35^**	**0.002**	**2.99 × 10^−22^**
	VS-MPR	**8.23 × 10^+10^**	**22,118.9**	**8.27 × 10^+11^**	**4.25 × 10^+8^**	**1.47 × 10^+32^**	**25.84**	**2.48 × 10^+19^**
	η² p	**0.304**	**0.145**	**0.325**	**0.256**	**0.644**	**0.073**	**0.707**
Time × Sex	F	1.19	0.10	3.73	2.78	**4.78**	1.55	0.01
	*p*	0.28	0.75	0.06	0.10	**0.03**	0.22	0.94
	VS-MPR	1.03	1.00	2.30	1.62	**3.46**	1.11	1.00
	η² p	0.01	6.60 × 10^−4^	0.02	0.02	**0.03**	0.01	6.45 × 10^−5^
Sex	F	2.40	**11.39**	0.46	2.15	0.01	2.73	0.02
(df:1)	*p*	0.12	**9.38 × 10^−4^**	0.50	0.15	0.93	0.10	0.90
	VS-MPR	1.42	**56.25**	1.00	1.31	1.00	1.59	1.00
	η² p	0.02	**0.07**	0.00	0.01	5.21 × 10^−5^	0.02	2.13 × 10^−4^

## Data Availability

De-identified data will be provided upon motivated request to the corresponding authors and approval of their Institutions.

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
