# Peer review of "In-Patient Multimodal Intensive Neurorehabilitation and Care Improve Motor and Non-Motor Functions in the Moderately Advanced Stages of Parkinson’s Disease: A Retrospective Study in a U.S. Facility"

_biomedicines, 2024, doi:10.3390/biomedicines12081658_

Round 1

Reviewer 1 Report

Comments and Suggestions for Authors

The study is very interesting and concerns a disability condition with increasing impact in our Western society. However, it is unclear why a study on a methodology of treatment already widely applied and proven effective in European studies should be repeated with a USA population.

The population study is large. The protocol used (multimodal, intensive, on inpatients) is very challenging economically: It would be appropriate to verify the cost and justify its appropriateness in terms of inpatients and outpatients.

In addition, the study has  some limitations, which moreover are pointed out by the Authors themselves. First of all, the study has no follow up so we do not know how much the results obteined with a very expensive demanding treatment are maintained even for a short time; it has not verification of  equal efficacy  in outpatient treatment, which would be less exxpensive; how long in fact should be the effective duration of the treatment (not more, not less) and how often it should be repeated.

I propose to develop the study  further by extending the review and the discussion  to the above considerations. 

Author Response

Comment Q1. The study is very interesting and concerns a disability condition with increasing impact in our Western society. However, it is unclear why a study on a methodology of treatment already widely applied and proven effective in European studies should be repeated with a USA population.

The population study is large. The protocol used (multimodal, intensive, on inpatients) is very challenging economically: It would be appropriate to verify the cost and justify its appropriateness in terms of inpatients and outpatients.

Answer A1. Thank you for this comment. As stated at the end of the introduction, (“ An initial validation of the European results of iMINC in the US system is warranted because of the differences about socio-economical context and type of health care system”) validation is needed in the USA, because of differences in many respects, including the fact that ours is a two-week program compared to a four-week European programs. We agree with the referee that establishing cost-effectiveness of the iMINC program is crucial for broader implementation. However, cost-effectiveness analysis can be done only on long follow-up studies. We are planning to do so in the next future about the population we are currently following up. This paper was geared at showing feasibility of an inpatient program in USA and the demonstration that the immediate outcomes are similar to those of the related European program. In terms of cost-effectiveness, here we would like to point out that: 1. the two-week program at Glen Cove Hospital has been entirely supported by Medicare; 2. the benefits of such programs show at least a significant improvement in the symptoms of the patients and their quality of life; 3. it provides direct (two-week respite) and indirect (patients’ improvements) for the caregivers. We now express these considerations in the discussion (see page 7 line 32-36):

“A crucial aspect for a broader implementation of the program is its cost-effectiveness, an aspect that can be settled only with long-term follow-up studies. So far, this two-week program has been approved and paid for by Medicare. Additional advantages include direct (a two-week respite period) and indirect (patients’ improvements) benefits for the caregivers.”

About a possible comparison with inpatient program: unfortunately, there is no outpatient program comparable to the Glen Cove Hospital program paid by Medicare. Comparison with current outpatient programs would be unfair. Indeed, the available outpatient (home-bound or community-based) programs usually are not comprehensive, do not last more than one hour per day, are not available every day, do not include transportation to an from facilities, etc.. A recent European a work -which is mentioned in the introduction and discussion- compared the effects of inpatient treatments similar to iMINC and those of PT outpatient programs (Ghilardi et al., 2024): it found that the subjects who underwent inpatient treatment had a better motor outcome over time (almost two years) than those who were managed by expert neurologists and participated in outpatient programs. Besides motor improvement, similar effects were seen for PDQ39, the number of falls, and hospitalization.  We are currently considering establishing, piloting, and testing the feasibility of outpatient programs that should have characteristics similar to the existing inpatient program at Glen Cove Hospital. This is no easy task overall for cost reasons. To expand this concept in the paper we have added on Page 7 (Lines 43-48) the following part:

“While such a comparison is not currently possible, a recent European a work studied the effects of inpatient treatments similar to iMINC to those of programs where patients managed by expert neurologists participated in enriched outpatient treatments. The results showed that outpatient programs produced a better motor outcome over almost two-year time, improved quality of life, and reduced the number of falls and hospitalization [27].”

Q2. In addition, the study has  some limitations, which moreover are pointed out by the Authors themselves. First of all, the study has no follow up so we do not know how much the results obteined with a very expensive demanding treatment are maintained even for a short time; it has not verification of  equal efficacy  in outpatient treatment, which would be less exxpensive; how long in fact should be the effective duration of the treatment (not more, not less) and how often it should be repeated.

I propose to develop the study  further by extending the review and the discussion  to the above considerations. 

A2. We agree with the referee that for the moment we do not have follow up studies of our young program at Glen Cove Hospital. However, there are European follow studies with a germane inpatient program showing long-term effects, as previously stated only in the introduction and now re-emphasized in the discussion on page 7 (lines 36-38):

“We also expect that our follow-up studies with a control group will still yield similar improvements, as germane European programs showed disease progression slowdown in the early stages [16] and increased independent functioning in the later stages [18–24].”

Reviewer 2 Report

Comments and Suggestions for Authors

Dear authors,

The topic of your paper is of interest, and the technical content is quite good. The topic is certainly worthy of investigation and is widely researched in the literature.

The introduction is well-written and outlines the background. Additionally, the methodology offers comprehensive details on the study design.

However, detailed statistical information in results, including effect sizes and confidence intervals are required. Additionally, a discussion of any potential outliers would be valuable.

There are also errors related to the formatting of the manuscript that need to be corrected to improve the readability and overall presentation of the document.

Author Response

Comment Q1. The topic of your paper is of interest, and the technical content is quite good. The topic is certainly worthy of investigation and is widely researched in the literature.

The introduction is well-written and outlines the background. Additionally, the methodology offers comprehensive details on the study design.

However, detailed statistical information in results, including effect sizes and confidence intervals are required. Additionally, a discussion of any potential outliers would be valuable.

A1. We thank you the referee for the positive comments. For the statistical question, we now specify in the methods that we have computed η² p  and Cohen’s d for ANOVAs and Welch tests, respectively. Confidence intervals (95% CI) have been reported for the post hoc Welch tests. Please notice that for ANOVAs, we computed Vovk-Sellke maximum p-ratio (VS-MPR) to assure that the absolute probability of a particular model is a good explanation of the observed data. We now clarify this in the text (see below and pg. 4, lines 25-33):

“… Bayes Factor approximation was also considered by computing the Vovk-Sellke maximum p-ratio (VS-MPR) (Sellke et al., 2001), a function of p-value that represents the lower bound of the Bayes Factor - favoring H0 to H1- for a wide range of different prior distributions. This also assures that the absolute probability of a particular model to be a good explanation of the observed data. VS-MPR values greater than 2.46 were considered significant. We also verified effect size by computing partial eta-squared (η² p), i.e., the % of the variance in the dependent variable attributable to a particular independent variable. Differences between men and women in terms of age and length of stay were analyzed with Welch tests for unequal samples with p value significance set at 0.05 and VS-MPR at 2.46. We also determined the effect size with Cohen’s d values together with 95% confidence intervals (CI).”

Q2. There are also errors related to the formatting of the manuscript that need to be corrected to improve the readability and overall presentation of the document.

A2. Thank you for pointing this out. We corrected all the formatting errors we found. Please let us know if there are still problems.

Reviewer 3 Report

Comments and Suggestions for Authors

Dear Author: 

This retrospective study investigates the effects of an in-patient Multimodal Intensive Neurorehabilitation and Care (iMINC) program on motor and non-motor functions in patients with moderately advanced Parkinson's disease (PD) at a US facility.There are some issues need to be clarified. 

1.         The study does not include a control group, which makes it difficult to attribute the observed improvements solely to the iMINC program.

2.         The study describes the multimodal intervention, but does not analyze the individual contributions of each component (e.g., physical therapy, occupational therapy, speech therapy). Future research should include a detailed analysis of the specific contributions of each component of the iMINC program.

3.         The study does not address the cost-effectiveness of the iMINC program, which is crucial for broader implementation.      Conduct a cost-effectiveness analysis to evaluate the financial feasibility of the iMINC program. This should consider the costs of the intervention relative to the benefits in terms of reduced hospitalizations, improved functionality, and enhanced quality of life.

Comments on the Quality of English Language

None

Author Response

Comment Q1. This retrospective study investigates the effects of an in-patient Multimodal Intensive Neurorehabilitation and Care (iMINC) program on motor and non-motor functions in patients with moderately advanced Parkinson's disease (PD) at a US facility.There are some issues need to be clarified. 

  1. The study does not include a control group, which makes it difficult to attribute the observed improvements solely to the iMINC program.

Answer 1. Thank you for this comment and we agree with the reviewer. In fact, in the original version we have touched upon it both in the introduction and the discussion.  We also mentioned that similar European studies with positive results included a control group (see for instance: first and third paragraph of the discussion as well as page 7 lines 28-30).

Q2. 2.         The study describes the multimodal intervention, but does not analyze the individual contributions of each component (e.g., physical therapy, occupational therapy, speech therapy). Future research should include a detailed analysis of the specific contributions of each component of the iMINC program.

A2. Thank you for this comment. In the previous version we approached this problem in different parts. For instance, on page 7 (lines 12-19) we stated that:

“In summary, all the components of  iMINC probably contribute to the beneficial effects observed in the patients: while goal-directed tasks may engage specific pathways that are affected by PD, aerobic exercise may provide a fundamental background for increasing plasticity and maintaining the effects of training; the environment and the care may boost reward system further enhancing plasticity and positive effects; increased attention to drug therapy may further improve the outcome. It is likely that synergy between all these interventions may produce a greater impact than the sum of the effects of iMINC’s individual components.”

Also that question was part of the open questions at the end of the discussion. To be more explicit we have now added an extra question (in red, see page 7, line 49-50) as follows: “Other questions include: which are the specific contributions of each component of the iMINC program? Which of the components or combination is more relevant?”

Q3. 3.         The study does not address the cost-effectiveness of the iMINC program, which is crucial for broader implementation.      Conduct a cost-effectiveness analysis to evaluate the financial feasibility of the iMINC program. This should consider the costs of the intervention relative to the benefits in terms of reduced hospitalizations, improved functionality, and enhanced quality of life.

A3. Thank you for this comment. We agree with the referee that establishing cost-effectiveness of the iMINC program is crucial for broader implementation. However, cost-effectiveness analysis can be done only on long follow-up studies. We are planning to do so in the next future about the population we are currently following up. This paper was geared at showing feasibility of an inpatient program in USA and the demonstration that the immediate outcomes are similar to those of the related European program. In terms of cost-effectiveness, here we would like to point out that: 1. the two-week program at Glen Cove Hospital has been entirely supported by Medicare; 2. the benefits of such programs show at least a significant improvement in the symptoms of the patients and their quality of life; 3. it provides direct (two-week respite) and indirect (patients’ improvements) for the caregivers. We now express these considerations in the discussion (see page 7 line 32-36):

“A crucial aspect for a broader implementation of the program is its cost-effectiveness, an aspect that can be settled only with long-term follow-up studies. So far, this two-week program has been approved and paid for by Medicare. Additional advantages include direct (a two-week respite period) and indirect (patients’ improvements) benefits for the caregivers.” 

Round 2

Reviewer 1 Report

Comments and Suggestions for Authors

I thank the Autohrs for the responses and I believe that now the new version can be published

Reviewer 3 Report

Comments and Suggestions for Authors

Dear Author: 

       This observational study has tried to improve upon the limitations of not having a control group. Besides, it points out future research should include a detailed analysis of the specific contributions of each component of the iMINC program. Finally, we look forward a crucial aspect for a broader implementation of the program is its cost-effectiveness. I suggest to accept and publish this article.

Comments on the Quality of English Language

None